# Detection of Oxacillin/Cefoxitin Resistance in *Staphylococcus aureus* Present in Recurrent Tonsillitis

**DOI:** 10.3390/microorganisms11030615

**Published:** 2023-02-28

**Authors:** Aline Cristine Magalhães Costa Messias, Aline Rodrigues Gama, Letícia Suriano de Almeida Prado, Paulo Alex Neves da Silva, Arlindo Rodrigues Galvão Filho, Clarimar José Coelho, Melissa Ameloti Gomes Avelino, José Daniel Gonçalves Vieira, Lilian Carla Carneiro

**Affiliations:** 1Institute of Tropical Pathology and Public Health, Federal University of Goiás, 235 Street, Goiânia 74605-020, GO, Brazil; 2Faculty of Medicine, Federal University of Goiás, 235 Street, Goiânia 74690-900, GO, Brazil; 3Engineering School, Pontifical Catholic University of Goiás, Avenue Universitária, Goiânia 74175-120, GO, Brazil

**Keywords:** bacteria, diagnostic imaging, children, antibiogram, contamination

## Abstract

Background: Recurrent tonsillitis is one of the most common diseases in childhood, caused many times by ß-lactam-resistant *S. aureus*. The objective of this study was to investigate an alternative method to identify resistance to oxacillin/cefoxitin in *S. aureus* from hospitalized children with recurrent tonsillitis. Methods: The samples of *S. aureus* came from patients with recurrent tonsillitis and were used in 16S rRNA sequencing and an antibiogram test for identification and verifying resistance, after which HSI methodology were applied for separation of *S. aureus* resistances. Results: The *S. aureus* isolated showed sensitivity to oxacillin/cefoxitin and the diagnostic images show a visual description of the resistance different groups formed, that may be related to sensitivity and resistance to oxacillin/cefoxitin, characterizing the MRSA *S. aureus*. Conclusions: Samples that showed phenotypic resistance to oxacillin/cefoxitin were clearly separated from samples that did not show this resistance. A PLS-DA model predicted the presence of resistance to oxacillin/cefoxitin in *S. aureus* samples and it was possible to observe the pixels classified as MRSA. The HSI was able to successfully discriminate samples in replicas that were sensitive and resistant, based on the calibration model it received.

## 1. Introduction

Recurrent tonsillitis is among the most common diseases in childhood, so, children are subjected to different antimicrobials as a form of treatment. However, the continuous and indiscriminate use of these drugs can cause insufficient efficacy, inducing tonsillectomy as an option for resolution of the infectious process. Many situations of inefficiency in the treatment of recurrent tonsillitis can be attributed to the presence of resistant bacteria colonizing the anatomical region [1].

The *Staphylococcus aureus* (*S. aureus*) is one of the pathogenic bacteria that causes recurrent tonsillitis. Factors such as the ability to acquire resistance and the presence of the clumping factor B (ClfB) protein favor the colonization of bacteria residing in the upper respiratory tract (RSV), in the anterior region of the nostrils, contributing to the development of tonsillitis [2,3].

A study shows that the nasal cavity, one of the main sites for the spread of bacteria in upper airways (UA), is persistently colonized by *S. aureus*, in 20% of the population [4]. It is estimated that 30% of individuals who develop infection with *S. aureus* in the nasopharynx are due to bacterial migration between sites [5]. Migration can reach the oropharynx and can be identified on the surface or in the nucleus of the tonsils [6].

For decades, antibacterials have become the main means of treating infectious diseases of bacterial origin. However, from 1950, the appearance of antimicrobial resistance began to be described in the literature, and the ineffectiveness of several treatments has resulted in deaths, due to the lack of antibacterials capable of eliminating multidrug-resistant bacteria [7].

Resistance to antibacterials, in cases of recurrent tonsillitis, is mainly related to amoxicillin, considered a drug of choice for the treatment of upper respiratory tract infections. From the 1970s, records of failure between 2% to 10% of amoxicillin therapy began to be published, the most likely cause being the production of the ß-lactamase enzyme, which inactivates the action of the antibacterial, a mechanism of common resistance related to *S. aureus* [8,9]. 

In the 1950s, the pharmaceutical industry made synthetic penicillins resistant to ß-lactamase, oxacillin, and methicillin available. These drugs have become alternatives to treatment for tonsillitis, caused by ß-lactam-resistant *S. aureus* [10]. After constant use of synthetic penicillins, *S. aureus* resistance to methicillin (MRSA—methicillin resistant *Staphylococcus aureus*) has been reported. Resistance was broadly identified to practically all ß-lactams: penicillin, methicillin, dicloxacillin, nafcillin, oxacillin, and cephalosporins [11].

Another form of resistance is due to the acquisition of the mecA gene, responsible for the modified production of the penicillin-binding protein (PBP), called PBP2a or PBP2’. Preventing the binding of ß-lactams to the target [12]. The mecA is inserted in a staphylococcal cassette, a mobile genetic element called SCCmec [13]. There are two basic genetic elements in its composition, the ccr and mec genetic complexes. Each element contains genes that, when inserted in the cassette, will promote specific resistance to various antibacterials and to heavy metals [14,15].

The identification of MRSA strains in the medical routine is based on the Antimicrobial Sensitivity Test (AST), also known as an antibiogram. To predict resistance to oxacillin mediated by the mecA gene, cefoxitin must be tested and the result reported to oxacillin [16]. The reference method applied to check the molecular resistance is conducted with the search for the mecA gene; this method is not performed in the hospital routine due to the high cost [17].

Bacterial resistance has become a challenge for public health, the alternative of blocking or preventing the spread of resistance is a difficult possibility, considering that the problem involves the human, animal, and environmental area [18]. A quick diagnosis of resistant strains can assist in the correct treatment and decrease the spread of this microorganism. In these cases, it would be essential to have an alternative method in the detection of these resistances [19].

Alternative method has been used as an analytical tool to aid in the agricultural, food, and pharmaceutical industry [20] and their identification of different bacterial genera isolated in drinking water [21].

Spectrophotometry methods have been used as a more agile process for phenotyping microorganisms [22]. One method that has stood out in such an approach is the use of hyperspectral images (HSI) [23]. In microbiology, HSIs have been applied in the characterization of bacterial colonies of medical importance, where analyses can be performed in a much shorter time, in comparison with conventional analysis methods [24]. In the same way, this technology has already been applied to environmental microbiology, with the identification of different bacterial genera isolated in drinking water [21].

An HSI has spectral and spatial information of a sample analyzed by combining conventional spectroscopic and digital images. Figure 1 shows a schematic representation of multiple bacteria culture in a Petri dish. The aforementioned combination is exemplified in Figure 1a by two spatial dimensions (x and y axes) with a third dimension referring to the wavelength (λ). Figure 1b shows two spectra referring to two different bacteria (red and blue). Different reflectance can be observed in the course of the wavelengths. Such a difference may be able to distinguish species of bacteria or even resistance to antimicrobials. The spectra can be named as spectral signatures of the respective samples of study [25] Some authors provides an example of this information [26,27].

There are still no records of the use of HSI in the detection of bacterial resistance, but in the study by [21], it occurred that when the technique was applied in differentiating bacteria, there was some variations in the spectrum in strains that belonged to the same genus and species. This factor can be justified by the presence or absence of some protein or even expression of resistance gene, that one bacterium could contain and the other not. It is suggested that with standardization of the spectral model, and using bacteria that contain resistance to antibacterials, it will be possible to apply a way of identifying the sensitivity profile from different spectral signals [21,28].

In this aforementioned context, the objectives of this study were to identify *S. aureus* from patients with recurrent tonsillitis, using 16S rRNA sequencing and investigating an alternative method for identifying oxacillin/cefoxitin resistance in these samples.

## 2. Materials and Methods

The following study is observational and cross-sectional. The samples were obtained between January to December, 2019. The approval protocol number of the Ethics Committee of the Hospital das Clínicas of the Federal University of Goiás was CAAE 84908818.3.00005083 (Ethics Committee of Hospital das Clínicas, from Federal University of Goiás).

### 2.1. Sampling

The samples of *S. aureus* used in the present study came from patients with recurrent tonsillitis, seen at an otorhinolaryngology outpatient clinic at Hospital das Clínicas, Universidade Federal de Goiás. The isolates were previously identified by phenotypic test based on the Anvisa (2013) and protocol of Koneman et al. (2008).

### 2.2. Amplification of the Coding Gene for the 16S rRNA Subunit

The bacterial isolates were identified by sequencing the 16S rRNA according to the in-house methodology described by Van Soolingen [29]. The genetic material was extracted and stored at −4 °C until preparation for sequencing. The conventional PCR product was obtained using the universal primers 27F (5′AGAGTTTGATCCTGGCTCAG-3′) and 1541R (5′- AAGGAGGTGATCCAGCC-3′), purified with the Agarose Extraction Kit (Cellco^®^, São Paulo, Brazil).

### 2.3. Sequencing the 16S rRNA Coding Region and Analyzing the Results

Sequencing was performed using primers 27F, 1541R, 926F (5′-AAACTYAAAKGAATTGACGG-3′), 530F (5′-TGACTGACTGAGTGCCAGCMGCCGCGG-3′), 519R (5′- GTNTTACNGCGGCKGCT-GNGGCGNGG-3′); the sequencer used was ABI 3500 (Applied Biosystems^®^ at CREBIO, Foster City, CA, USA). The sequences obtained were analyzed using the Codon Code Aligner software (Codon Code Corporation—demo version) and compared for homology, with the GenBank NCBI database using BLASTn. The sequences obtained were deposited at GenBank NCBI (Table 1).

### 2.4. Sample Preparation for Obtaining Images

The samples were sown on nutrient agar and after 24 h, they were resuspended in sterile saline (NaCl 0.85%), standardized with turbidity compatible with the McFarland 0.5 scale (1 × 10^6^ CFU/mL). A swab was moistened in this bacterial suspension and seeded on nutrient agar by the sweeping technique. Two plates were prepared for each bacterial strain of *S. aureus*. All bacterial plaques were incubated for 24 h at 35 °C.

### 2.5. Acquisition of Hyperspectral Images

The selected *S. aureus* samples were replicated twice, including a total of four samples positioned at random. HSI were obtained using a SisuChema workstation (Specim, Spectral Imaging Ltd., Oulu, Finland) by means of a camera capable of capturing short wave infrared images (Short Wave Infrared—SWIR) with Chemadaq software version 3.62.183.19.

The system consists of an image spectrograph coupled to a mercury-cadmium-tellurium detector (HgCdTe) of 2-D matrix with a light source of quartz halogen lamps mounted in a reflective box. A 50 mm high magnification lens with a spatial resolution of 0.30 µm was used to capture images at a frame rate of 100 Hz at a 3.0 ms exposure with a spectral range of 920–2514 nm and a resolution 6–7 nm 256 × 320 pixels, and a pixel depth of 14 bits/pixel were obtained. 

After capturing the test image, the MRSA 2 bacteria that were replicated were placed under the same conditions at room temperature. Dark and white internal reference standards were used for the calibration image and to correct the variation in the sample illumination.

### 2.6. Principal Component Analysis

Principal Component Analysis (PCA) consists of a multivariate analysis technique that can be used to analyze interrelationships among other variables, applied as a centered average of all images, including the background (Jolliffe, 2002). It was applied with a centralized average to all our images.

The obtained images were automatically corrected for white and dark references and converted to pseudo-absorption (A/D converter counts to absorb) using the multivariate analysis software Evince version 2.4.0 (UmBio AB, Umeå, Sweden) (Manley et al., 2011). PCA reduces the dimensionality of large data sets by decomposing interrelated variables into a new set of coordinates (PCs) [30,31].

We used the score image and the scatter density plots to remove pixels corresponding to the background, dead pixels, and border effects [26,27], revealing the variation between the samples. Some mathematical pretreatment methods such as normal variation and derivatives minimize variability unrelated to the chemical composition of the powder that was investigated for spectral preprocessing [32].

The method that best separated the differences from relevant images was chosen. The optimal number of PCs was determined by excluding all PCs that did not significantly increase the Q2 Ycum value. The regions at the beginning and at the end of the near infrared (NIR) spectrum are generally not very informative. The modeling of these regions hides useful chemical information and hence, the need to investigate these and remove them from the model so that the chemical differences are clear.

In this study, wavelengths ≤ 996 nm at the beginning of the spectra (920–996 nm) did not contain differentiated chemical information and were excluded from the data set, which resulted in better pixel classification. The resulting image was then evaluated for chemical differences and similarities in order to differentiate samples of *S. aureus* positive for oxacillin/cefoxitin resistance.

### 2.7. Partial Least Squares Discriminatory Analysis

Partial Least Squares Discriminatory Analysis (PLS-DA) is a technique that recognizes patterns that correlates variation in the spectral data matrix X to an independent variable Y, being categorical and resulting in the discrimination or classification of samples. This is a super-classified classification technique, since the data are assigned to previous knowledge for an effective prediction of the adhesion of groups in new samples [24]. 

In this study, variable Y was generated through test sample 1 and 2, and the PCA with a standard image previously calibrated with methicillin-resistant *Staphylococcus aureus* (register CCBH 6089) (MRSA). In total, there were six samples in the image of the PCA score, two of which were a calibration set while four were the others which formed the test set. The cross-validation of the model was performed using a random selection method.

Seven iterations of cross-validation were performed as standard settings in the software. PLS factors were added to the model and the optimal number was determined by excluding factors that did not significantly increase the value of Q2 Ycum in the model. The PLS-DA score image (Y image) was then used for the adhesion/prediction of resistance (Min Y cut: 0.5; maximum Y cut: 1.5) of the test set and MRSA 2 samples.

## 3. Results

### 3.1. Sequencing the 16S rRNA

We conducted the sequencing of *S. aureus* using the gene 16S rRNA. After confirming the *S. aureus* by sequencing, the sequence was analyzed by NCBI and deposited on Gen- Bank. In Table 2 are shown sequences from GenBank that presented higher identity with these study samples.

Sequencing of the 16S ribosomal RNA region demonstrates 92.44% identity with test sample 1 and 99.69% identity with test sample 2 from the database (NCBI).

### 3.2. Oxacillin/Cefoxitin Resistance and Sensitivity

Samples 1 and 2 of *S. aureus* showed sensitivity to oxacillin/cefoxitin. 

### 3.3. Image Analysis Using PCA

The exploratory analysis was carried out in two ways, with a spectrum of NIR images and a graph of PCA points. Figure 2 shows the NIR spectra of MRSA 2 samples in replicas and samples 1 and 2 of this study. Spectral patterns may not have distinctive characteristics that can be used to differentiate resistances and hence, the need for PCA that provides visual graphics such as punctuation images and scatter plots to observe clearer differences.

The interactive score image and scatter plots were colored according to the PC score values. The score image is an amplitude graph where similar colors represent similar score values and vice versa. Samples that showed phenotypic resistance to oxacillin/cefoxitin were clearly separated from samples that did not show this resistance.

Figure 3 shows a visual description of the resistance groups, using amplitude and density from individual pixels. It is possible to observe different groups formed that may be related to sensitivity and resistance to oxacillin/cefoxitin.

The observation supports the results of the scoring image, where the maximum variance modeled over the PC1 (75.2%) was demonstrated to mainly differentiate the sensitive samples. Distinctions were observed in the resistant samples along the PC2 in the dispersion plot.

To investigate the differences observed in sensitivity and resistance in the scoring images, load line plots of the first vector (P1) were constructed for the model (Figure 4). The load plots show the region between 900–2357 nm as carrying discriminatory information regarding resistance and sensitivity. The presence of positive spikes (peaks) and negative charges (troughs) were recorded in this region.

### 3.4. Image Analysis Using PLS-DA

A PLS-DA model was developed to predict the presence of resistance to oxacillin/cefoxitin in *S. aureus* samples. This model was constructed using the two duplicates of the MRSA 2 samples as a calibration set and the two replicates of the samples obtained from the patients. Figure 5a shows the HSI of two replicates of the MRSA 2 samples as the calibration set, together with two replicates of the oxacillin/cefoxitin sensitive and resistant samples as the test set. The prediction result for calibration and test sets are shown in Figure 5b. The green-colored pixels were classified as MRSA, and the red-colored ones were classified as no class. As expected, most of the pixels in the calibration samples were identified as MRSA 2. The same behavior was observed in the test samples, where the model classified most of the pixels in the samples that contained resistances as MRSA 2. The capacity of the model in the detection of resistance is once again validated with the sensitive samples of the test set. It is possible to observe that the pixels were classified as MRSA.

PLS-DA shows the calibration set with test set; both were used to train the equipment. The predicted class provided the results shown in Figure 5b, where it shows the resistance with a different color (green) while the background or some unknown areas were classified as “unidentified” (red). The test materials were in accordance with the previous knowledge about the resistance and sensitivity of the samples, based on the color code.

The HSI was able to successfully discriminate samples in replicas that were sensitive and resistant, based on the calibration model it received.

## 4. Discussion

Studies have shown that *S. aureus* is prevalently in research that evaluates the pathogens involved in recurrent tonsillitis in pediatric patients [2]. In the present study, sequencing the 16S rRNA region confirmed the presence of *S. aureus* in the pathogenicity of recurrent tonsillitis. The ability of this microorganism to acquire and transmit resistance is a factor that raises the importance of identifying it [33]. 

A study carried out in Milan collected swab samples from the anterior nostrils and pharynx of pediatric patients, who would undergo surgery classified as clean; 138 (35.1%) children had positive screening for *S. aureus*. MRSA was identified in 40 (29% of positive cases for *S. aureus*) children; having this microorganism as a colonizer makes the risk of developing serious infections caused by it greater, however, the study did not characterize whether MRSA was associated with the community or hospital [34].

Before the 1990s, most strains of MRSA were reported as HA-MRSA (hospital associated methicillin resistant *Staphylococcus aureus*). Since the 1990s, strains of CA-MRSA have been increasingly found among groups of patients with no apparent connection to hospitals. A large percentage of CA-MRSA strains are already known to be pediatric, which poses a threat to the latent spread of highly virulent MRSA [34]. 

Another study which was carried out in Iran investigated the presence of MRSA in children who attended kindergarten, without any known factor for MRSA colonization; in total, 354 children were analyzed, among them, 20 children (5.8%) tested positive for colonization of community-acquired methicillin-resistant *Staphylococcus aureus* (CA-MRSA) [35,36]. 

Chmielowiec-Korzeniowska [37] demonstrated the presence of MRSA in healthy individuals, stating that having the bacterium as a common colonizer in the microbiota is one of the main therapeutic and epidemiological problems. Colonization favors the maintenance of the microorganism in a population, and asymptomatic transport in a chronic way acts as a reservoir of the strain, favoring its propagation in the environment [37].

The prevalence of methicillin-resistant *S. aureus* is increasing over time, with records of its importance in head and neck infections, where treatment becomes more difficult due to bacterial virulence factors. Among the existing infections, we can highlight tonsillitis [38]. In the present study, there was an indication of the presence of MRSA in one of the pediatric patients suffering from recurrent tonsillitis identified, using the antibiogram test, showing resistance to oxacillin/cefoxitin.

The phenotypic antibiogram takes an average of 3 to 5 days to perform, and this time may vary because the bacteria have different growth times. Before and after the test, there are different guidelines regarding the incubation time for reading and interpreting the results, and this variation can be according to the bacteria being tested. The automated antibiogram is faster than the conventional one, around two days, however, there is still a limitation on the time of bacterial growth for its performance [39,40]. 

With the results obtained in this study, we suggest that HSI analysis can differentiate bacteria between resistant and sensitive, however, to be possible, this characterization is based on the proposed technology, and it is necessary to build databases that are capable of covering the resistance of the existing bacterial species, as was developed in this work. Based on the level of differentiation obtained, we suggest that this study will initiate new research that may make it possible to apply this technology for both identification and definition of the resistance profile, in addition to extending to different areas of microbiology.

## 5. Conclusions

According to the sequencing data, the presence of *Staphylococcus aureus* was confirmed in the analyzed samples and with sensitivity to oxacillin/cefoxitin.

The diagnostic image was carried out in two ways, with a spectrum of NIR images and a graph of PCA points. Samples that showed phenotypic resistance to oxacillin/cefoxitin were clearly separated from samples that did not show this resistance. The load plots confirmed the discriminatory information regarding resistance and sensitivity.

A PLS-DA model predicted the presence of resistance to oxacillin/cefoxitin in *S. aureus* samples and it was possible to observe the pixels classified as MRSA. The HSI was able to successfully discriminate samples in replicas that were sensitive and resistant based on the calibration model it received.

## Figures and Tables

**Figure 1 microorganisms-11-00615-f001:**
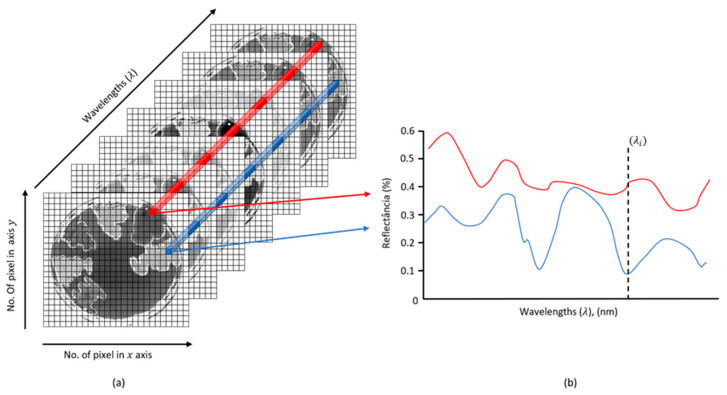
Schematic representation of an (**a**) HSI of multiple bacteria culture in a Petri dish and (**b**) a spectral signature of two different pixels on the hypercube image.

**Figure 2 microorganisms-11-00615-f002:**
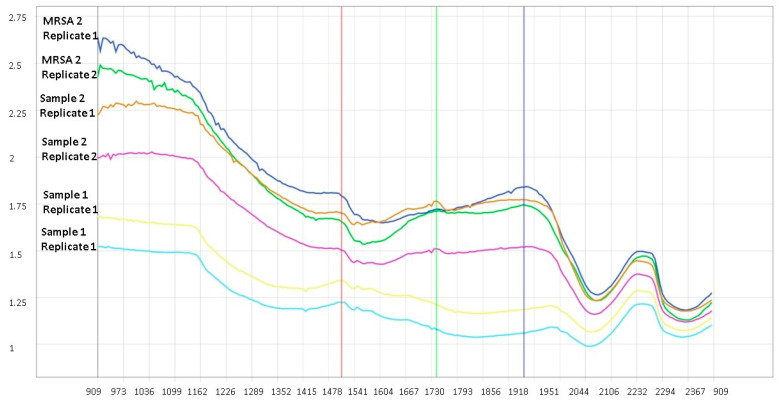
Average spectrum of HSI of MRSA samples and test samples. Blue color refers to MRSA 2/replicate 1; green color refers to MRSA 2/replicate 2.

**Figure 3 microorganisms-11-00615-f003:**
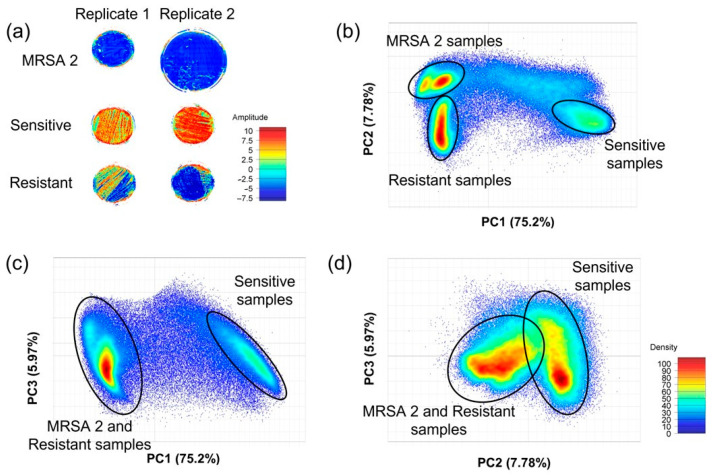
PCA image score showing the amplitude of the signal represented by the blue color in the image, where there is the presence of resistance to oxacillin/cefoxitin (**a**). Corresponding scoring graph (PC1 × PC2) shows groups of distinct pixels colored according to the density color scale (**b**); the same occurs on the scoring graph (PC1 × PC3) where all the resistant samples were gathered at the same point (**c**), and on the scoring graph (PC2 × PC3), the samples that contained resistance showed higher intensity than the sensitive sample (**d**).

**Figure 4 microorganisms-11-00615-f004:**
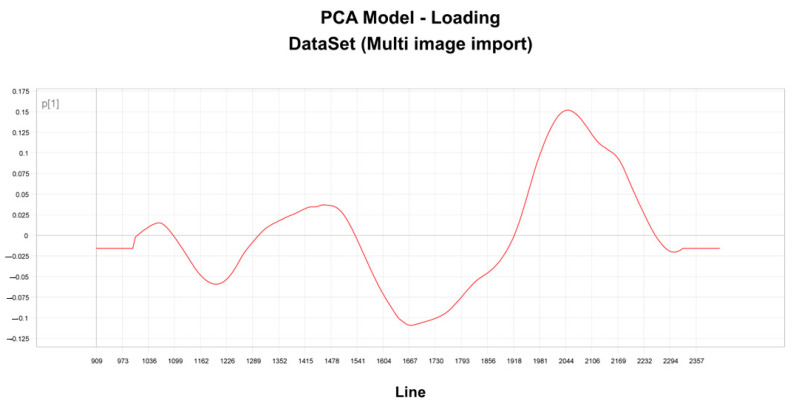
Load line of vector P1 for the score image showing variables responsible for the separation of resistance and sensitivity.

**Figure 5 microorganisms-11-00615-f005:**
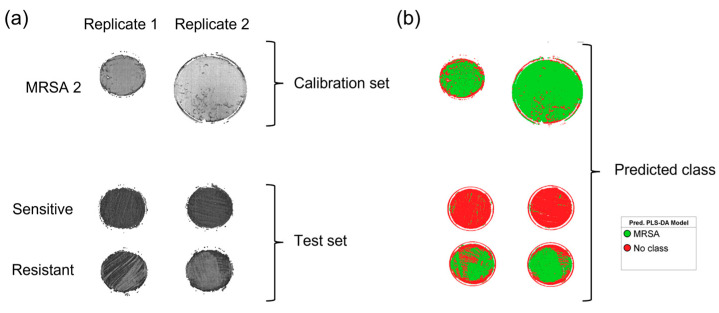
PLS-DA image showing calibration samples used to build the model and test samples that were excluded for model validation (**a**). The predictions show test samples assigned to resistance that correspond to the generated images (**b**).

**Table 1 microorganisms-11-00615-t001:** Sequence of 16 S rRNA, product of sequencing in this work.

Microorganisms	Deposit Code
1 *Staphylococcus aureus*	*
2 *Staphylococcus aureus*	MW480543

* Sequence not accepted for deposit due to insufficient size.

**Table 2 microorganisms-11-00615-t002:** Identity of *Staphylococcus aureus* deposited on database, comparing with the Staphylococcus aureus of this study.

NCBI Sample	GenBank Register	Sample Test 1 Identity %	Sample Test 2 Identity %
*Staphylococcus aureus* strain S33 R 16S ribosomal RNA	NR_037007	92.44%	99.69%

## Data Availability

Not applicable.

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
