# Peer review of "Detection of Oxacillin/Cefoxitin Resistance in Staphylococcus aureus Present in Recurrent Tonsillitis"

_microorganisms, 2023, doi:10.3390/microorganisms11030615_

Round 1
Reviewer 1 Report
article ``Detection of oxacillin/cephoxitin resistance in Staphylococcus aureus present in recurrent tonsillitis'' is an interesting, methodologically correctly written text. with clear results, excellently presented to the reading public. A discussion follows the results with appropriate references. excellently written manuscript
Author Response
reviewer 1 made no suggestions for changes.
Reviewer 2 Report
The authors have investigated an alternative method for identifying oxacillin/cefoxitin resistance of S.aureus strains from patients with recurrent tonsillitis. They used imaging-based assays to distinguish between resistant and sensitive strains. The work provides a novel way of differentiating various strain types. Overall experiments are well performed and data is well presented. However, the sample numbers used in the study are not that significant.
Minor comments
1. Reframe the abstract to focus on the main objective and important findings instead of dividing it into different sections.
2. Give a rationale for the experiments and the conclusions in the result section.
3. There are many grammatical mistakes which need to be addressed.
Author Response
Response to Reviewer 2 Comments
Point 1: Reframe the abstract to focus on the main objective and important findings instead of dividing it into different sections.
Response 1: The abstract has been redrafted as requested by the reviewer. Changes are highlighted in yellow.
Point 2: Give a rationale for the experiments and the conclusions in the result section.
Response 2: A short justification has been added in the RESULTS section. Changes are highlighted in yellow.
Point 3: There are many grammatical mistakes which need to be addressed.
Response 3: Spelling errors have already been proofread and corrected. Changes are highlighted in yellow.
Response to Reviewer 1 and 3 Comments
Reviewer 1
Point 1: article ``Detection of oxacillin/cephoxitin resistance in Staphylococcus aureus present in recurrent tonsillitis'' is an interesting, methodologically correctly written text. with clear results, excellently presented to the reading public. A discussion follows the results with appropriate references. excellently written manuscript.
Response 1: no suggestions for changes.
Reviewer 3
Point 1: According to the sequencing data, the presence of Staphylococcus aureus was confirmed in the analyzed samples and with sensitivity to oxacillin / cefoxitin.
The diagnostic image was carried out in two ways, with a spectrum of NIR images and a graph of PCA points. Samples that showed phenotypic resistance to oxacillin / cefoxitin were clearly separated from samples that did not show this resistance. The load plots confirmed the discriminatory information regarding resistance and sensitivity.
A PLS-DA model predicted the presence of resistance to oxacillin/cephoxitin in S. aureus samples and it was possible to observe the pixels classified as MRSA. The HSI was able to successfully discriminate samples in replicas that were sensitive and resistant, based on the calibration model it received.
Response 1: no suggestions for changes.
Please see the attachment.

Reviewer 3 Report
According to the sequencing data, the presence of Staphylococcus aureus was confirmed in the analyzed samples and with sensitivity to oxacillin / cefoxitin.
The diagnostic image was carried out in two ways, with a spectrum of NIR images and a graph of PCA points. Samples that showed phenotypic resistance to oxacillin / cefoxitin were clearly separated from samples that did not show this resistance. The load plots confirmed the discriminatory information regarding resistance and sensitivity.
A PLS-DA model predicted the presence of resistance to oxacillin/cephoxitin in S. aureus samples and it was possible to observe the pixels classified as MRSA. The HSI was able to successfully discriminate samples in replicas that were sensitive and resistant, based on the calibration model it received.

Author Response
reviewer 3 made no suggestions for changes.